# Dyslipidemia among adult HIV patients on antiretroviral therapy and its association with age and body mass index in Ethiopia: A systematic review and meta-analysis

**Abebe Muche Belete**[1]*, **Daniel Molla Melese**[1], **Bekalu Bewket**[2], **Belachew Tegegne**[2], **Wondimeneh Shibabaw Shiferaw**[3,4], **Yared Asmare Aynalem**[5], **Adisu Asefa**[6], **Taklo Simeneh Yazie**[7]

1 Department of Biomedical Science, Asrat Weldyes Health Science Campus, Debre Berhan University, Debre Berhan, Ethiopia, 2 Department of Nursing, College of Health Science, Injibara University, Injibara, Ethiopia, 3 UQ Centre for Clinical Research, Faculty of Medicine, The University of Queensland, Herston, Australia, 4 Department of Nursing, Asrat Weldeyes Health Science Campus, Debre Berhan University, Debre Berhan, Ethiopia, 5 Department of Nursing, University of Alberta, Edmonton, Canada, 6 Department of Non communicable disease, Armauer Hansen Research Institute, Addis Ababa, Ethiopia, 7 Department of Pharmacy, College of Health Science, Debre Tabor University, Debre Tabor, Ethiopia

* abebemuche3@gmail.com

## Abstract

### Introduction

Dyslipidemia is a common public health problem in people living with human immunodeficiency virus (HIV) who are receiving antiretroviral therapy and increases the risk of cardiovascular disease. Although evidence indicates that the prevalence of dyslipidemia is high, estimated pooled data are not well documented. Therefore, we aimed to estimate the pooled prevalence of dyslipidemia in adult people living with HIV receiving antiretroviral therapy in Ethiopia.

### Method

We conducted a systematic review and meta-analysis of the literature. The following databases and grey literature were searched: PubMed, WorldCat, ScienceDirect, DOAG, African Journals Online, Google Scholar, and African Index Medicine. We included all comparative epidemiological studies that reported the prevalence of high concentration of total cholesterol, triglycerides, and low density lipoprotein, and low concentration of high density lipoprotein cholesterol that were published between January 2003 and July 2023. The random effects model was used to pool the outcome of interest. Additionally, subgrouping, sensitivity analyses, and funnel plots were performed. R software Version 4.2.1 was used for statistical analysis.

### Result

Seventeen studies with a total of 3929 participants were included in the meta-analysis. The pooled prevalence of dyslipidemia, high total cholesterol, high triglyceride, elevated level of

**Data Availability Statement:** All relevant data are within the manuscript and its Supporting information files.

**Funding:** The author(s) received no specific funding for this work.

**Competing interests:** The authors have declared that no competing interests exist.

**Abbreviations:** ART, Antiretroviral Therapy; BMI, Body Mass Index; CI, confidence interval; CVD, Cardiovascular disease; HDL-c, High-Density Lipoprotein-cholesterol; HIV, Human Immune Deficiency Virus; LDL-c, Low-Density Lipoprotein-cholesterol; OR, Odds ratio; PLHIV, People living with HIV; TC, Total Cholesterol; TG, Triglyceride.

low density lipoprotein and low level of high density lipoprotein cholesterol were 69.32% (95% CI: 63.33, 74.72), 39.78% (95%CI: 32.12, 47.96), 40.32% (95%CI: 34.56, 46.36), 28.58% (95%CI: 21.81, 36.46), and 36.17% (95%CI: 28.82, 44.24), respectively. Age and body mass index were associated with high total cholesterol, triglyceride, and low-density lipoprotein cholesterol levels.

## Conclusion

The authors concluded that the prevalence of dyslipidemia in Ethiopia is high in people living with HIV receiving antiretroviral therapy. Early detection of dyslipidemia and its integration into treatment are essential for preventing cardiovascular disease.

## Trial registration

Protocol registered with PROSPERO (CRD42023440125).

## Introduction

Dyslipidemia is an abnormality in a single or combination of lipid profiles such as high total cholesterol (TC), low-density lipoprotein cholesterol (LDL-C), triglycerides (TGs), and low high-density lipoprotein cholesterol (HDL-C) levels [1]. It is a common public health problem among people living with HIV (PLHIV) and is mainly associated with antiretroviral therapy (ART) [2]. It is a major risk factor for cardiovascular disease (CVD) [3, 4], which can lead to heart attack and stroke.

The prevalence of dyslipidemia varies across the countries. For instance, it was found 40.2% in Nigeria [5], 51% in Cameron [6], 86.6% in Eritrea [7], 75% in Kenya [8], and 63.9% in Ethiopia. There are reviews of dyslipidemia in Sub-Sharan Africa [9–11] and in the globe; among those in the globe, two were with meta-analysis [12, 13] but one without [14].

There is no review of prevalence studies of dyslipidemia in Ethiopia, and only a few Ethiopian studies are cited in African and global reviews. Additionally, the prevalence of dyslipidemia varied within regions, highlighting the need for this review. Thus, our study systematically reviewed the existing evidence on the prevalence of dyslipidemia among adult PLHIV receiving ART. We also conducted a meta-analysis of relevant studies to develop clinically transferable strategies. This finding provides insight into the public health problems of dyslipidemia in PLHIV receiving ART and supports policymakers' in the prevention, detection and control strategies.

## Materials and methods

We followed the guidelines and instructions in the updated protocol of Preferred Reporting Items for Systematic Review and Meta-analysis (PRISMA) [15] (S1 Table). The study protocol was prospectively registered with PROSPERO (CRD42023440125). The review question was formed according to the Joanna Briggs Institute (JBI) Reviewer's Manual for prevalence studies and the CoCoPop (Condition, Context and Population) [16]. In the current review, dyslipidemia, Adult HIV patients receiving ART, and Ethiopia are considered as the condition, population, and context, respectively. Therefore, the review's research questions were: (i) What is the prevalence of dyslipidemia among PLHIV receiving ART in Ethiopia? (ii) What is the prevalence of high LDL-c, TC, TG, and low HDL-c among PLHIV receiving ART in

Ethiopia? and what is the association of dyslipidemia with age, and body mass index (BMI) among PLHIV receiving ART in Ethiopia?

## Eligibility criteria

Following the CoCoPop framework, the eligibility of studies was determined as follows:

**Population.**   We included studies involving adults (18 years or older) living with HIV receiving ART. We excluded studies involving children and adolescents with HIV.

**Condition.**   We considered studies that reported the prevalence of dyslipidemia, high concentrations of TC, LDL-c, and TGs and low (decreased) concentrations of HDL-c.

**Context.**   We included studies conducted in the community, and hospital- that reported the prevalence of dyslipidemia and/or high TC, TGs, and LDL-c and a low HDL-c. Only studies conducted in Ethiopia were included in the systematic review and meta-analysis.

**Type of study.**   Observational studies were included.

## Information source

We used the Peer Review of Electronic Search Strategies (PRESS) methodology for systematic reviews in our search strategy [17]. The process involves two researchers, a requester and reviewer, both of whom are skilled in the search for bibliographic databases. First, the requester filled out the pertinent information in the updated PRESS 2015 Guideline Assessment Form for the "primary" search strategy and sent it to a reviewer. The reviewer reviewed the search strategy using the PRESS 2015 Evidence-Based Checklist (available in Table 1 in McGowan et al. 2016 [17]). The search strategy was reviewed by the full team. The databases used were PubMed, African Journals Online, African Index Medicus, DOAG, WorldCat, ScienceDirect and AfroLib. In addition, the reference lists in the papers were examined for relevant papers. Papers were also searched from grey literature, such as Google Scholar. Literature searching was performed between 20 June 2022 and 12 July 2022, and updated on 23 August 2023. Studies published between June 2003 and August 2023 were included. Only the studies published in English were included.

**1.1 Search strategies.**   The initial key words used were HIV, ART, dyslipidemia, and Ethiopia. Search terms for HIV included "HIV" OR "human immunodeficiency virus" OR "AIDS" OR "acquired immunodeficiency syndrome." Search terms of ART included "highly active antiretroviral therapy," "HAART," OR "ART," OR "antiretroviral therapy," "protease inhibitor," OR "NNRTI" OR "NRTI," OR "Integrase inhibitor," OR "PI" OR "PIs" OR "lopinavir" OR "ritonavir" OR "lamivudine" OR "zidovudine" OR "stavudine" OR "nevirapine" OR "efavirenz" OR "tenofovir" OR "emtricitabine" OR "atazanavir" OR "darunavir." Search terms for dyslipidemia included "dyslipidemias" OR "hypercholesterolemia" OR "hypertriglyceridemia" OR "total cholesterol" OR "TC" OR "cholesterol blood level" OR "triglyceride" OR "TG" OR "high-density lipoprotein cholesterol" OR "HDL-c" OR low-density lipoprotein cholesterol" OR "LDL-c" OR "non-HDL" AND "Ethiopia."

**1.2. Selection of studies.**   A search strategy was used to obtain titles and abstracts of studies that may be relevant to the review. We exported search results into Endnote version 8.1 for bibliographic citation management and excluded duplicate references. Two authors (WSS and AMB) independently screened the titles and abstracts, and discarded studies that were not applicable. The other two authors (YAA and TSY) independently screened full texts for inclusion while recording the reasons for the exclusion of ineligible studies. Any disagreements were resolved through discussion or, if required, a third review author was consulted.

**1.3. Methodological quality assessment.**   The included studies were evaluated for methodological quality using the Newcastle-Ottawa Scale (NOS) [18]. In each included study, we

**Table 1. Study characteristics of included studies.**

| Author | Pub. Year | Region | Study design | Sample size | Study regimen | dyslipidemia | TC | | TG | | LDL | | HDL | | |
|---|---|---|---|---|---|---|---|---|---|---|---|---|---|---|---|
| | | | | | | | | | | | | | | cutoff | |
| | | | | | | Prev (%) | Prev (%) | Cutoff in mg/dl | Prev (%) | Cutoff in mg/dl | Prev (%) | Cutoff in mg/dl | Prev (%) | Male in mg/dl | Female in mg/dl |
| Kemal A et al. [23] | 2020 | Addis Ababa | Cross sectional | 353 | All ART | 74.8 | 45.9 | ≥200 | 28.9 | ≥150 | 31.2 | ≥130 | 35.7 | <40 | <50 |
| Gebrie A et al. [24] | 2020 | Amhara | Cross sectional | 407 | All ART | 63.9 | 34.2 | ≥200 | 39.6 | ≥150 | 27.8 | ≥130 | 21.1 | <40 | <50 |
| Belay E et al. [25] | 2014 | Addis Ababa | Cross sectional | 70 | All ART | 80 | 45.7 | ≥200 | 40 | ≥150 | 40 | ≥130 | 22.9 | <40 | <40 |
| Yazie TS [26] | 2020 | Addis Ababa | Prospective cohort | 63 | TDF-based | 77.8 | 42.9 | ≥200 | 31.7 | ≥150 | 41.3 | ≥130 | 41.3 | <40 | <40 |
| Tadewos A et al. [27] | 2012 | Sidama | Cross sectional | 113 | All ART | 82.3 | 43 | ≥200 | 55.8 | ≥150 | 33.6 | ≥130 | 43.4 | <40 | <40 |
| Berhane T et al. [8] | 2012 | Oromia | Cross sectional | 313 | | | 48.2 | 6.7 | ≥240 | 18.2 | ≥200 | 6.9 | ≥160 | 32.6 | <40 | <40 |
| Agete T and Demissie A [28] | 2015 | Sidama | Prospective cohort | 78 | EFV Vs NVP | 74.4 | 35.8 | ≥200 | 60 | ≥150 | 26.9 | ≥130 | 30.7 | <40 | <40 |
| Fiseha T et al. [29] | 2021 | Amhara | Cross sectional | 392 | All ART | 59.9 | 47.3 | ≥200 | 30.9 | ≥150 | 29.6 | ≥130 | 19.4 | <40 | <40 |
| Bayenes HW et al. [30] | 2014 | Addis Ababa | Cross sectional | 114 | All ART | 73.7 | 50 | ≥200 | 59.6 | ≥150 | 48.3 | ≥130 | 43.8 | <40 | <40 |
| Tesfaye Y et al. [2] | 2014 | Sidama | Cross sectional | 188 | All ART | - | - | - | 45.2 | ≥150 | | | 53.1 | <40 | <50 |
| Abebe M et al. [34] | 2014 | Oromia | Cross sectional | 126 | All ART | - | 42.1 | ≥200 | 46.8 | ≥150 | 23 | ≥130mg/dl | 50.8 | <40 | <40 |
| Muche BA et al. [35] | 2021 | Addis Ababa | Cross sectional | 180 | EFV and ATV/r | - | 28.9 | >200 | 39.4 | >200 | 50 | ≥129 | 45.6 | <40 | <40 |
| Ataro Z and Ashenafi W [36] | 2020 | Hareri | Cross sectional | 375 | All | | - | | 41.9 | ≥150 | - | - | 64.5 | <40 | <50 |
| Bune GT et al. [37] | 2020 | SNNP | Cross sectional | 422 | All | - | - | | 42.9 | ≥150 | - | - | 30.8 | <40 | <50 |
| Tilahun A et al. [31] | 2022 | Amhara | Cross-sectional | 114 | All | 73.6 | 50 | >200 | 59.6 | ≥150 | 48.2 | ≥130 | 43.8 | <40 | <50 |
| Assefa A et al. [32] | 2023 | Addis Ababa | Cross-sectional | 288 | All | 55.2 | 22.5 | >200 | 18.7 | ≥150 | 4 | ≥130 | 48.6 | <40 | <50 |
| Woldeyes E et al. [33] | 2022 | Addis Ababa | Cross-sectional | 333 | All | 69.4 | 52.3 | >200 | 37.2 | ≥150 | 40 | ≥130 | 10.6 | <40 | <50 |

assessed the representativeness, response rate, and method, comparability of the subjects and appropriateness of the statistical analysis used. These items were marked out of ten. Then, studies scored five and above were included in the final analysis. Two authors (BB and BT) assessed the quality of the studies using the above criteria (S2 Table). Any disagreements were resolved through discussion.

**1.4. Data extraction.** We used the Joanna Briggs Institute (JBI) extraction form for prevalence and incidence studies available in Munn et al. [16]. Two authors (WSS and AMB)

performed the data extraction. The following characteristics were extracted from the included studies: author, publication year, region, study design, sample size, outcome, outcome reported, cutoff value, and associated factors. Disagreements were resolved by consensus or discussion.

**1.5. Assessment of risk of bias.** Each included study was evaluated using the Hoy risk of bias assessment tool for reporting prevalence data [19]. The Hoy score is marked out of ten and a value of 8–10 indicated low bias, 5–7 moderate bias, and ≤4 high bias (S3 Table). Two authors (YAA and AMB) independently assessed the risk of bias.

**1.6. Heterogeneity and publication bias.** We used Cochran's $Q$ and $I^2$ statistics to measure heterogeneity among the studies included in each analysis [20]. Higgins et al. suggested using an $I^2$ value $\geq 75\%$ to indicate high heterogeneity. We performed a subgroup analysis based on region and study design. Sensitivity analysis was also conducted for the effect of each study on the overall prevalence. Egger's test was used to assess publication bias [21].

**1.7. Statistical analysis.** Both meta-analysis and narrative synthesis were used. DerSimonian–Laird random-effects model [22] was used to generate the pooled prevalence of dyslipidemia, high TC, TGs, LDL-c, and low HDL-c. The pooled effect size (i.e., prevalence) was weighted, and a 95% confidence interval (CI) was generated. Additionally, the pooled effect size (i.e., odds ratio) was generated with its 95% CI. All analysis are displayed in the form of a forest plot. R software version 4.2.1 was used.

## Result

### Description of studies

**Result of the search.** A database search identified 163 studies. After removing duplicates, 63 studies remained. We excluded 33 studies at the stage of the abstract and title screening. We reviewed the remaining 30 studies for further details and excluded 13 studies for several reasons (Fig 1) (S4 Table).

**Included studies.** Seventeen studies with 3929 participants were included in the analysis. The study and patient characteristics, including the study setting, outcome report, study design, and sample size, are presented in Table 1. We identified 15 cross-sectional and 2 prospective cohort studies. Studies of 12 with 2638 participants, 14 consisting of 2928 participants, and 17 having 3929 participants were included in the pooled meta-analysis of dyslipidemia [8, 23–33], both TC and LDL-c [8, 23–35], and both TGs and HDL-c [2, 8, 23–37], respectively.

### Risk of bias

See S3 Table for the Hoy score.

### Prevalence of dyslipidemia

The present meta-analysis revealed that the overall prevalence of dyslipidemia in adult PLHIV receiving ART was 69.32% (95% CI: 63.33, 74.72), $I^2 = 90\%$, p < 0.01 (Fig 2). This showed substantial variation among the studies. We then applied a subgroup analysis on the study design and region. Accordingly, the pooled prevalence of dyslipidemia among cross-sectional studies was 68.1% [95% CI: 61.63, 73.94], $I^2 = 91\%$, p < 0.001. Whereas among prospective studies, it was 76.04% (95%CI: 59.3, 87.36), p = 0.64, $I^2 = 0\%$. Moreover, studies conducted in the Addis Ababa region showed that the pooled prevalence of dyslipidemia was 71.28% (95%CI: 64.45, 77.26), p = 0.74, $I^2 = 87\%$. Whereas in the Amhara region, it was 65.47% (95%CI: 55.53, 74.22), p = 0.25, $I^2 = 72\%$.

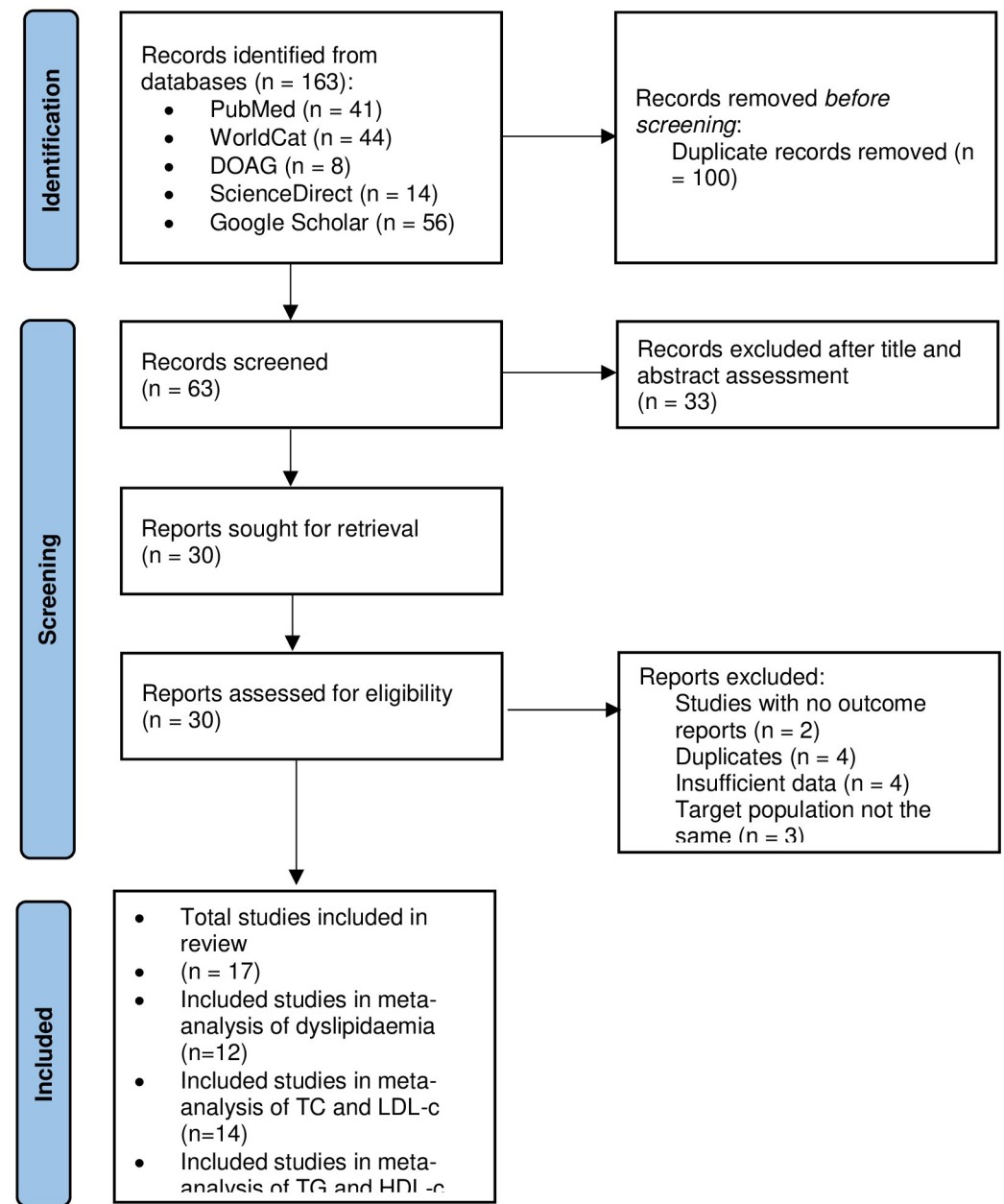

**Fig 1. PRISMA 2020 flow diagram summarizing the selection process of studies included in the systematic review and meta-analysis.**

## Prevalence of high TC

The overall pooled prevalence of high TC among PLHIV receiving ART was 39.78% (95%CI: 32.12, 47.96), $I^2$ = 94%, p<0.01 (Fig 3). Subgroup analysis was conducted on the region and study design. Accordingly, the pooled prevalence of high TC among studies conducted in the Addis Ababa and Amhara region were 44.59% (95% CI: 35.76, 53.78), $I^2$ = 93%), p = <0.01, and 43.45% (95% CI: 29.97, 57.97), $I^2$ = 89%, p = <0.01, respectively. Additionally, the prevalence of TC among cross-sectional studies was 37.24% (95%CI: 29.33, 45.90), $I^2$ = 94%,

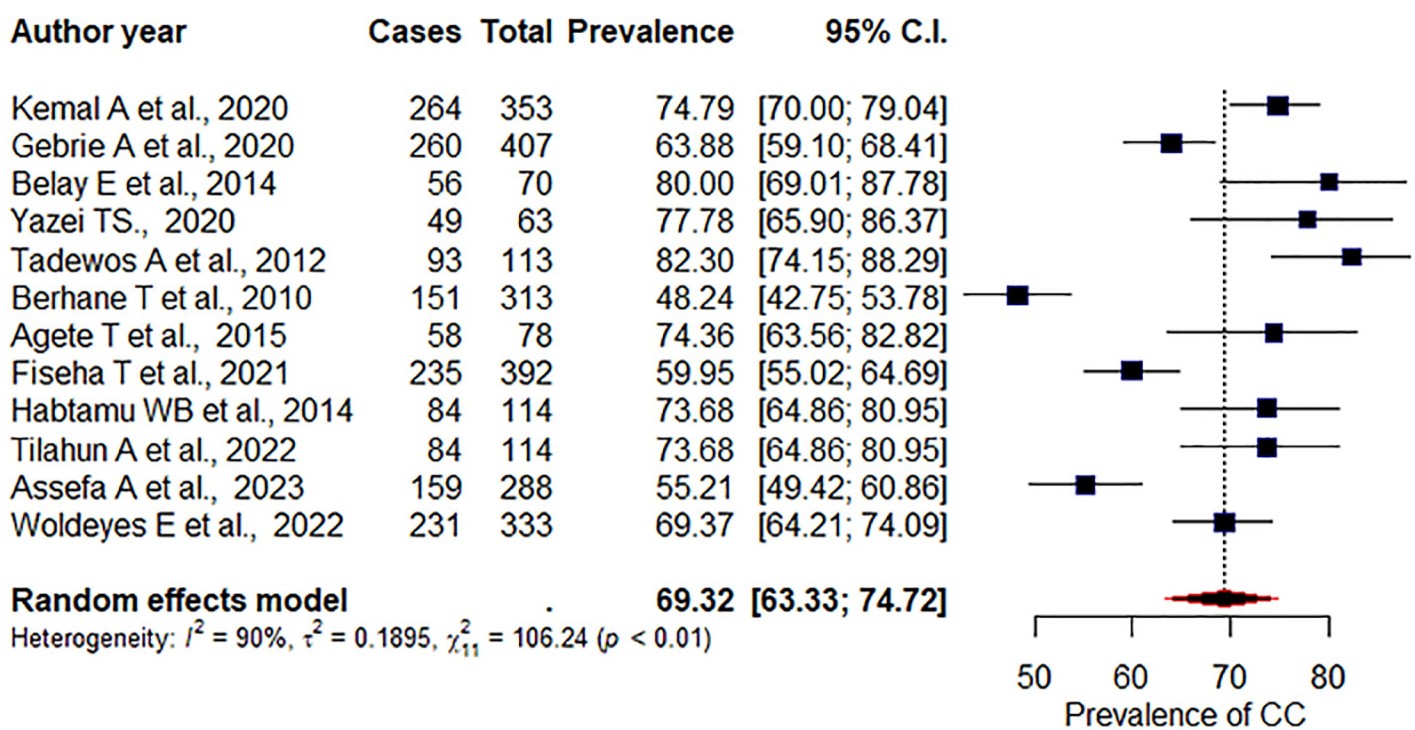

**Fig 2. Pooled prevalence of dyslipidemia.**

p = <0.001. We also used Egger's test to assess publication bias. Accordingly, the test results were not significant for publication bias (p = 0.57).

## Prevalence of high TGs

The overall pooled prevalence of high TGs among adult PLHIV receiving ART was 40.32% (95%CI: 34.56, 46.36), p = <0.01, $I^2$ = 92% (Fig 4). In sub group analysis, the pooled proportion of high TGs among studies conducted in Addis Ababa region and in Sidama were found to be 36.95% (95% CI: 29.08, 45.58), p = 0.01, $I^2$ = 91%, and 55.59% (95%CI: 41, 69.27), $I^2$ = 81%, p = <0.01, respectively. According to the study design, the pooled prevalence of high TGs in cross-sectional studies was 39.28% (95%CI: 33.33, 45.57), $I^2$ = 92%, p <0.01. Whereas in those prospective studies, we found that 49.57% (95%CI: 31.47, 67.77), $I^2$ = 94%, p<0.01. We did Egger's test to see publication bias. Accordingly, the test is not significant for publication bias (p = 0.14).

In the current studies, we performed pooled prevalence of high TGs among the following regimen: Tenofovir (TDF)-, Zidovudine (AZT)-, Stavudine (D4T)-, Efavirenz (EFV)-, and Nevirapine (NVP)-based regimen. The overall pooled prevalence of high TGs among regimens based-TDF-, AZT, D4T, EFV, and NVP were 34.61% [(95%CI: 22.97, 48.44), p = 0.19, $I^2$ = 40%],43.69% [(95%CI: 31.99, 56.15), p = 0.01, $I^2$ = 73%], 41.39% [(95%CI: 28.68, 55.36), p = 0.01, $I^2$ = 73%], 42.53% [(31.05, 54.88), p<0.01, $I^2$ = 76%], and 42.72% [(95%CI: 31.28, 55.01), p<0.01, $I^2$ = 81%], with the respective order.

## Prevalence of high LDL-c

The overall pooled prevalence of high LDL-c among adult PLHIV receiving ART was 28.58% (95%CI: 21.81, 36.46), p = <0.01, $I^2$ = 94% (Fig 5). Having subgroup analysis on region, the

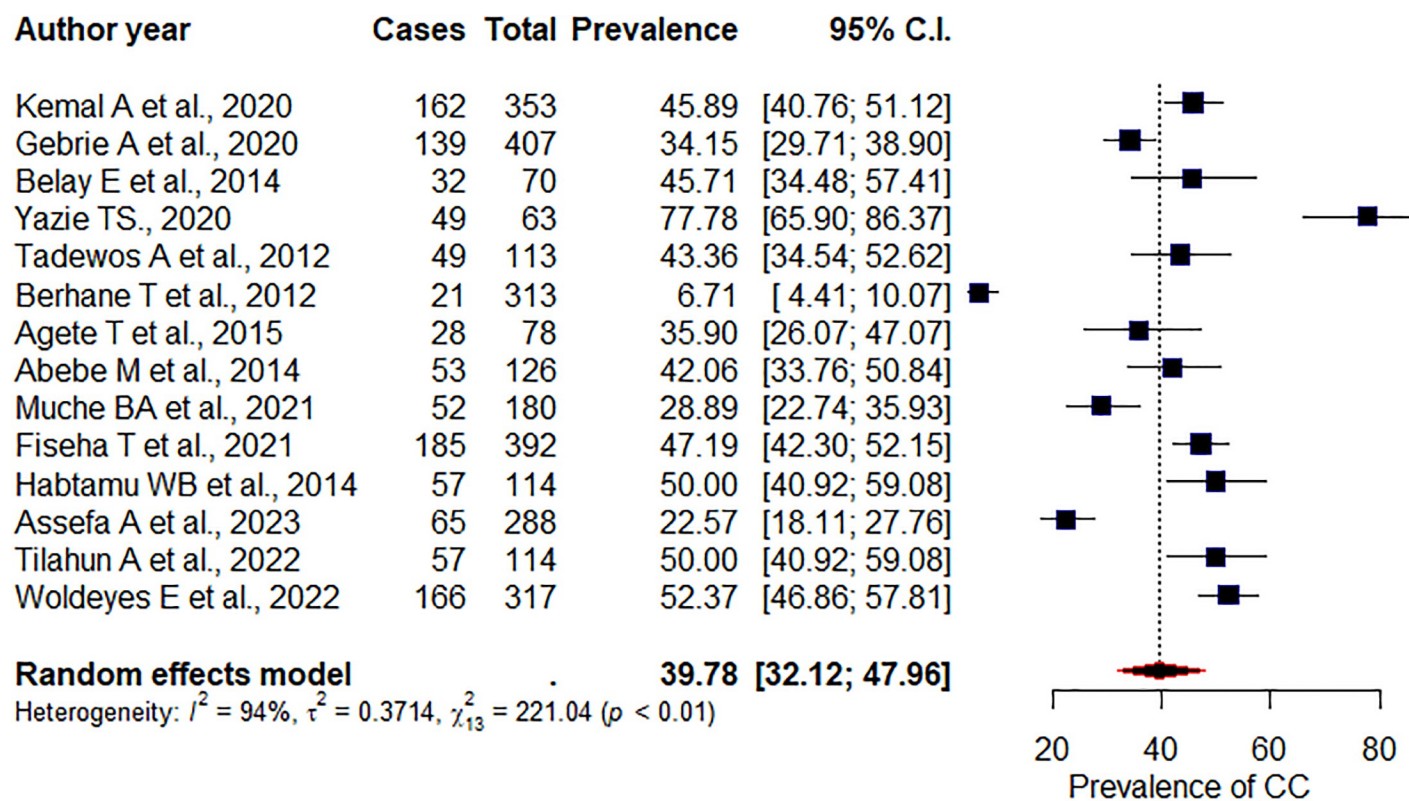

| Author year | Cases | Total | Prevalence | 95% C.I. |
|---|---|---|---|---|
| Kemal A et al., 2020 | 162 | 353 | 45.89 | [40.76; 51.12] |
| Gebrie A et al., 2020 | 139 | 407 | 34.15 | [29.71; 38.90] |
| Belay E et al., 2014 | 32 | 70 | 45.71 | [34.48; 57.41] |
| Yazie TS., 2020 | 49 | 63 | 77.78 | [65.90; 86.37] |
| Tadewos A et al., 2012 | 49 | 113 | 43.36 | [34.54; 52.62] |
| Berhane T et al., 2012 | 21 | 313 | 6.71 | [ 4.41; 10.07] |
| Agete T et al., 2015 | 28 | 78 | 35.90 | [26.07; 47.07] |
| Abebe M et al., 2014 | 53 | 126 | 42.06 | [33.76; 50.84] |
| Muche BA et al., 2021 | 52 | 180 | 28.89 | [22.74; 35.93] |
| Fiseha T et al., 2021 | 185 | 392 | 47.19 | [42.30; 52.15] |
| Habtamu WB et al., 2014 | 57 | 114 | 50.00 | [40.92; 59.08] |
| Assefa A et al., 2023 | 65 | 288 | 22.57 | [18.11; 27.76] |
| Tilahun A et al., 2022 | 57 | 114 | 50.00 | [40.92; 59.08] |
| Woldeyes E et al., 2022 | 166 | 317 | 52.37 | [46.86; 57.81] |
| **Random effects model** | | . | **39.78** | **[32.12; 47.96]** |

Heterogeneity: $I^2 = 94\%$, $\tau^2 = 0.3714$, $\chi^2_{13} = 221.04$ ($p < 0.01$)

**Fig 3. Pooled prevalence of high total cholesterol concentration.**

pooled prevalence of high LDL-c levels among studies carried out in the Amhara region was 34.48% (95%CI: 20.74, 51.41), p = 0.57, $I^2$ = 93%, in Addis Ababa 30.4% (95%CI: 21.95, 40.42), p< 0.01, $I^2$ = 94%. According to the study design, the pooled prevalence of high LDL-c levels among prospective cohort studies was found to be 49.37% (95%CI: 26.63, 72.38), p = 0.52, $I^2$ = 94%. Whereas studies of cross sectional showed 29.21% (95%CI: 21.83, 37.89), $I^2$ = 95%, p<0.01. We did Egger's test to see publication bias. Accordingly, the test is not significant for publication bias (p = 0.3).

The overall pooled prevalence for high LDL-c among AZT-based regimen was 30.1% (95% CI: 25.2, 35.49), p = 0.53, $I^2$ = 0%; among D4T-based regimen was 27.17% (95%CI: 21.16, 34.15), p = 0.58, $I^2$ = 0%; among EFV-based regimen was 28.31% (95%CI: 23.63, 33.51), p = 0.38, $I^2$ = 0%. And among NVP-based regimen was 32.03% (95%CI: 27.49, 36.94), p = 0.44, $I^2$ = 0%.

## Prevalence of low HDL-c

The overall pooled prevalence of low HDL-c levels was 36.17% (95%CI: 28.82, 44.24), p<0.01, $I^2$ = 96% (Fig 6). According to sub group analysis, the pooled prevalence of low HDL-c levels was 25.84% (95%CI: 14.67, 41.38), p = 0.18, $I^2$ = 94%, and 35.88% (95%CI: 26.5, 46.48), p<0.01, $I^2$ = 94%, among studies carried out in the Amhara region, and in the Addis Ababa region, respectively. The prevalence of low HDL-c among prospective cohort study design was 35.82% (95%CI: 16.69, 60.87), p = 0.2, $I^2$ = 40%, whereas it was 36.22% (95%CI: 28.35, 44.9),

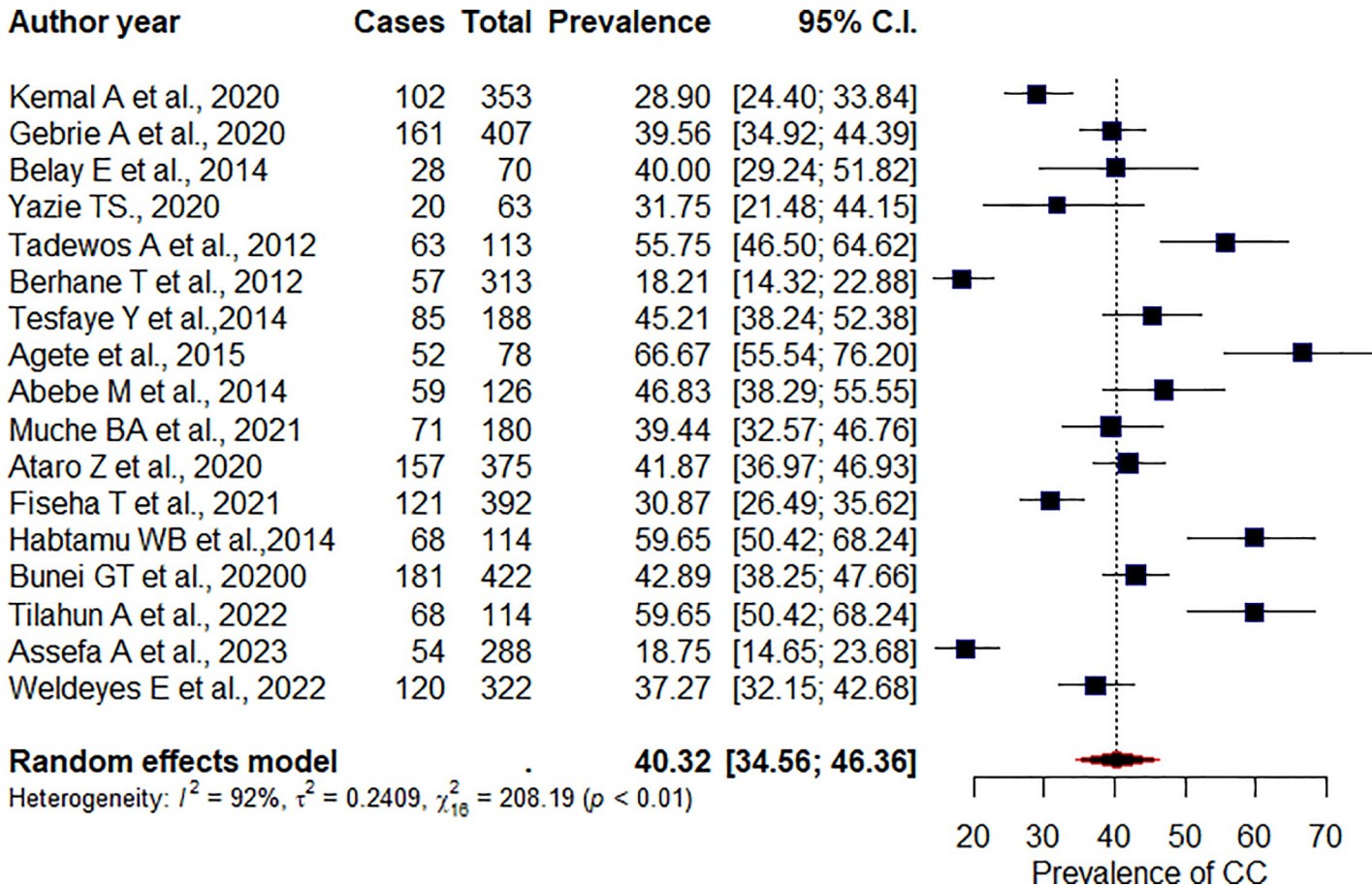

**Fig 4. Pooled prevalence of high triglyceride.**

p<0.01, $I^2$ = 96% among cross sectional studies. We did Egger's test to see publication bias. Accordingly, the test is not significant for publication bias (p = 0.74).

The overall pooled prevalence of low HDLC-c among TDF, AZT, D4T, EFV, and NVP-based regimen were 26.46% [(95%CI: 13.06, 46.28), p<0.01, $I^2$ = 84%], 32.84% [(95%CI: 18.62, 51.09), p<0.01, $I^2$ = 86%], 36.1 [(95CI: 20.16, 55.83), p<0.01, $I^2$ = 83%], 36.44% [(95%CI: 21.27, 54.88), p<0.01, $I^2$ = 91%], and 29.8 [(95%CI: 16.62, 47.47), p<0.01, $I^2$ = 86%], respectively.

### Factors associated with total cholesterol

In the current analysis, age and BMI were significantly associated with high TC as shown in Fig 7. The pooled effect of two studies [28, 29] revealed that age > 40 years was associated with high TC (OR = 2.6; 95%CI: 1.82, 3.71), $I^2$ = 0.0%. Moreover, the pooled effect of four studies [26, 27, 29, 30] showed that body mass index ≥25 was statistical associated with TC (OR = 2.71; 95%CI: 1.75, 4.2), $I^2$ = 0.0%. Furthermore, two studies in Amhara region reported the association of TC and CD4 counts but they used different cut off points that hinders us to do pooled effect analysis. Fiseha T et al. depicted that CD4 count less than 200 was significantly associated with TC [29], whereas Muche AB et al. found the significant association of CD4 count less than 455 and TC [35].

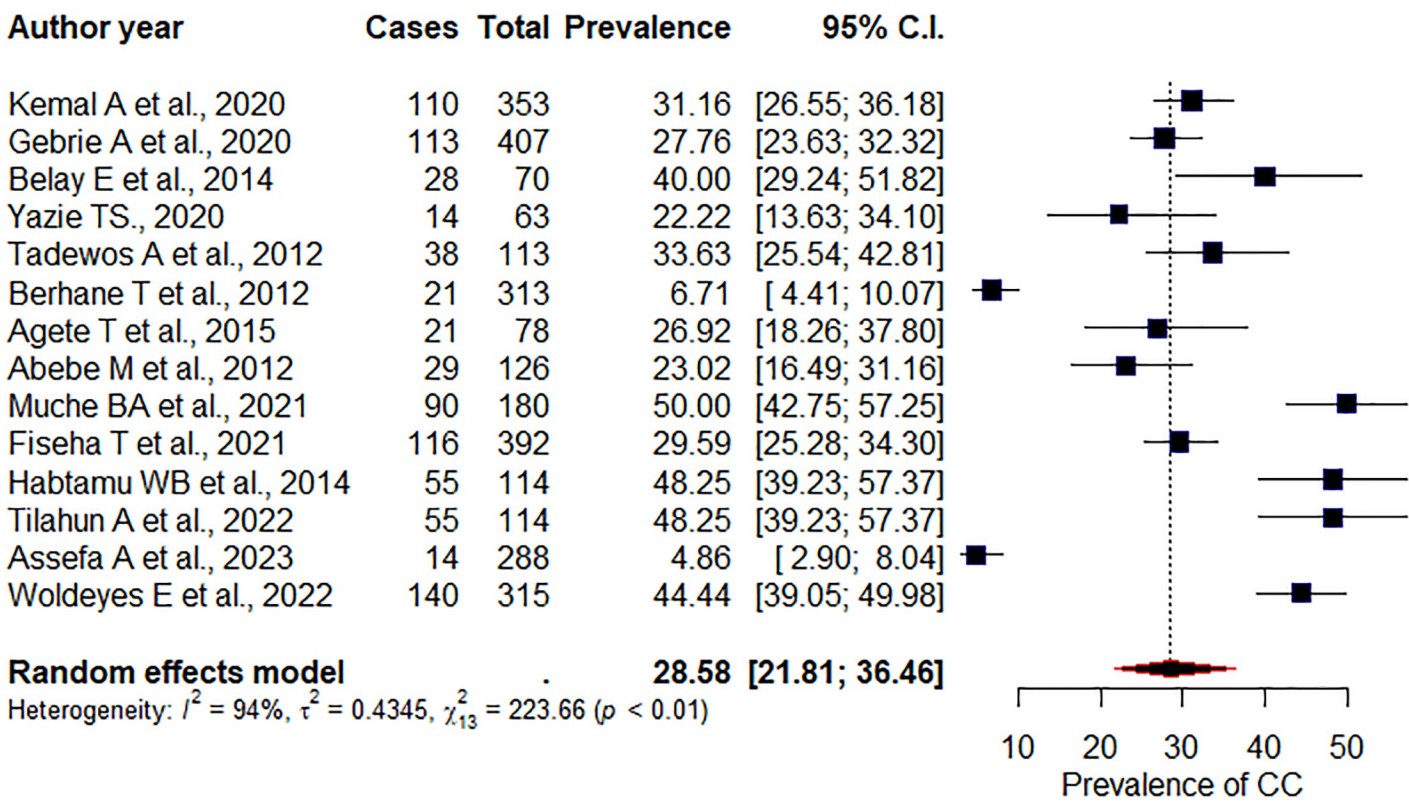

**Fig 5. Pooled prevalence of high LDL-c concentration.**

## Factors associated with triglyceride

In the present analysis, age and BMI were associated with high TGs as shown in Fig 8. Accordingly, the pooled effect of two studies [29, 30] showed that age > 40 years was statistically associated with high TG (OR = 2.77; 95%CI: 1.8, 4.28), $I^2$ = 0.0%. Furthermore, the pooled effect of three studies [27, 29, 30] showed that BMI ≥25 was associated with high TGs (OR = 3.59, 95% CI: 2.01, 6.41), $I^2$ = 0.0%.

## Factors associated with LDL-c

Again, age and BMI were associated with high LDL-c. Two studies [27, 29] were reported on the association of age and LDL-c. When we pooled them, age was statistically associated with high LDL-c (OR = 3.34, 95%CI: 2.02, 5.52), $I^2$ = 0.0%. Furthermore, the pooled effect of four studies [27, 29, 30, 35] showed that BMI ≥ 25 was associated with elevated levels of LDL-c (OR = 3.27, 95%CI: 2.15, 4.96), $I^2$ = 0.0% (Fig 9).

**Factors associated with dyslipidemia.** In our review, a study conducted in Amhara region by Gebrie et al. found the significant association of substance use (AOR = 13.38, 95% CI: 1.082–165.39), and drug therapy (AOR = 0.22, 95%CI: 0.07–0.63) for another disease with dyslipidemia [24]. In addition, a study conducted in Addis Ababa by Yazie TS reported a significant association between having cancer as a comorbidity and dyslipidemia [26]. However, we did not show the pooled odds ratio of factors as being reported in a single study. Moreover, we did not access studies in our literature search, which reported the association of CD4, opportunistic infections, and viral load with dyslipidemia or any lipid abnormalities.

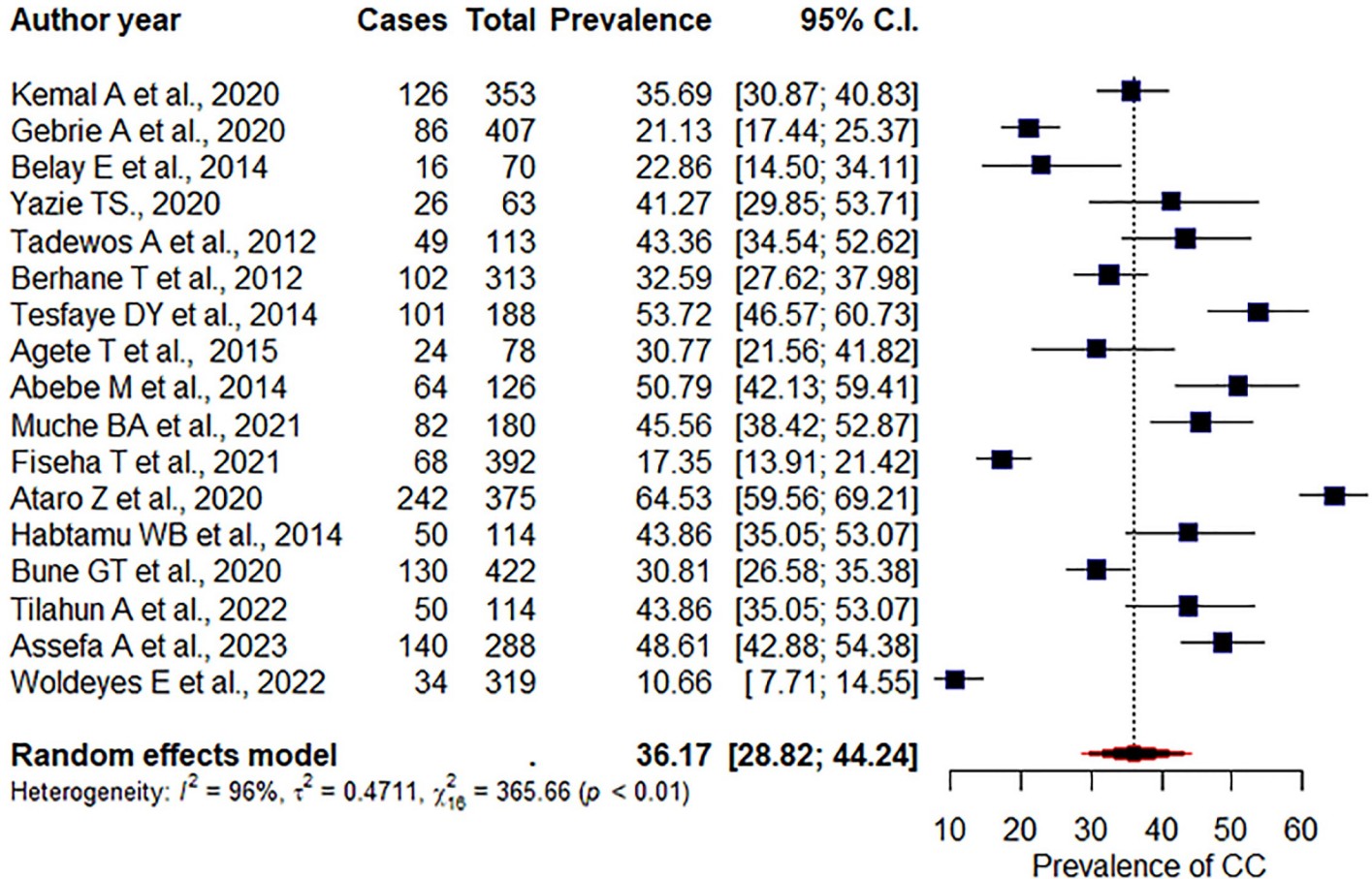

| Author year | Cases | Total | Prevalence | 95% C.I. |
|---|---|---|---|---|
| Kemal A et al., 2020 | 126 | 353 | 35.69 | [30.87; 40.83] |
| Gebrie A et al., 2020 | 86 | 407 | 21.13 | [17.44; 25.37] |
| Belay E et al., 2014 | 16 | 70 | 22.86 | [14.50; 34.11] |
| Yazie TS., 2020 | 26 | 63 | 41.27 | [29.85; 53.71] |
| Tadewos A et al., 2012 | 49 | 113 | 43.36 | [34.54; 52.62] |
| Berhane T et al., 2012 | 102 | 313 | 32.59 | [27.62; 37.98] |
| Tesfaye DY et al., 2014 | 101 | 188 | 53.72 | [46.57; 60.73] |
| Agete T et al., 2015 | 24 | 78 | 30.77 | [21.56; 41.82] |
| Abebe M et al., 2014 | 64 | 126 | 50.79 | [42.13; 59.41] |
| Muche BA et al., 2021 | 82 | 180 | 45.56 | [38.42; 52.87] |
| Fiseha T et al., 2021 | 68 | 392 | 17.35 | [13.91; 21.42] |
| Ataro Z et al., 2020 | 242 | 375 | 64.53 | [59.56; 69.21] |
| Habtamu WB et al., 2014 | 50 | 114 | 43.86 | [35.05; 53.07] |
| Bune GT et al., 2020 | 130 | 422 | 30.81 | [26.58; 35.38] |
| Tilahun A et al., 2022 | 50 | 114 | 43.86 | [35.05; 53.07] |
| Assefa A et al., 2023 | 140 | 288 | 48.61 | [42.88; 54.38] |
| Woldeyes E et al., 2022 | 34 | 319 | 10.66 | [7.71; 14.55] |
| **Random effects model** | | | **36.17** | **[28.82; 44.24]** |

Heterogeneity: $I^2 = 96\%$, $\tau^2 = 0.4711$, $\chi^2_{16} = 365.66$ ($p < 0.01$)

Prevalence of CC

**Fig 6. Pooled prevalence of HDL-c.**

## Discussion

We conducted this review as no data on pooled prevalence of dyslipidemia in Ethiopians PLHIV receiving ART. In our reviews for prevalence of dyslipidemia, the only regions with data were Addis Ababa (n = 7), Amhara regional state (n = 3), Sidama regional state (n = 3), Southern nation nationalities of people (n = 1), Oromia (n = 2), and Harari (n = 1). Thus, from the 12 regions of the country, only 6 regions had any study of prevalence of dyslipidemia or lipid profile. There were major regional variations in the prevalence of dyslipidemia, with much of the burden seen in Addis Ababa.

We intentionally loosened our criteria for inclusion and exclusion in order to obtain a rough estimate of the prevalence in Ethiopia. We believe that we have an imperfect yet useful estimate of at least general hospital patients for dyslipidemia prevalence of about 69.32%.

Our finding was higher than Marina G et al. outcome of 39.5% in the global review of dyslipidemia [12]. Additionally, a meta-analysis conducted in Sub-Saharan countries showed that ART use was associated with high TC, TG, LDL-c, and low HDL-c [10]. Furthermore, other studies also found higher levels of serum lipid profiles in PLHIV receiving ART [11]. The high prevalence of dyslipidemia among PLHIV who received ART highlights the need for regular monitoring and management of lipid levels in this population. The pattern of lipid abnormalities among PLHIV receiving ART is complex, and it differs from one regimen to the other.

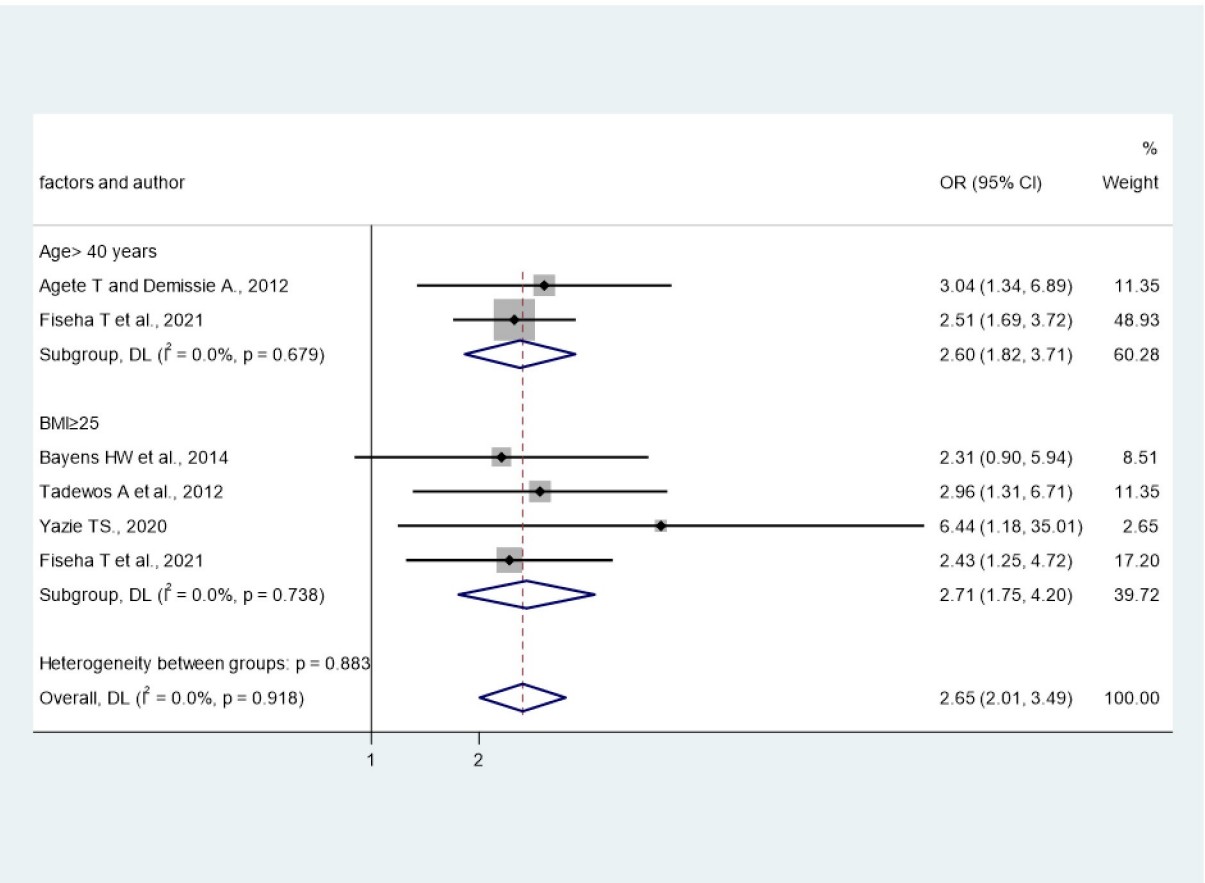

**Fig 7. The effects of age and BMI on total cholesterol.**

The overall pooled prevalence of high TC among adult PLHIV receiving ART was 39.78% with the higher prevalence that was observed among studies conducted in Addis Ababa region (44.59%) though in the African review of Noubiap JJ et al., prevalence of elevated TC was 26.2% [9]. This figure showed that Ethiopian PLHIV face particular problem. In the global review of Nduka C et al. [13] PLHIV receiving ART had elevated levels of TC as compared to naïve HIV patients.

The overall pooled prevalence of high TGs among adult PLHIV was 41.32%. This figure was higher than a review of African 24% [9]. No significant difference in the prevalence between Zidovudine (AZT), Stavudine (D4T), and Tenofovir Disoproxil Fumarate (TDF)-based regimen. However, with higher prevalence among AZT-based regimen as compared to D4T and TDF. Several studies have indicated that TDF may have a more beneficial effect on triglyceride levels compared to AZT and D4T in both treatment naïve and treatment-experienced individuals. Replacing stavudine with tenofovir could be a beneficial approach to enhance the lipid profile of patients with dyslipidemia, especially in terms of reducing triglycerides. Due to the negative impact of tenofovir on bone and kidney health, a modified version called (tenofovir alafenamide (TAF) has been developed. TAF does not affect lipid parameters. When patients switch from tenofovir to TAF, their LDL, HDL, and triglyceride levels increase. Conversely, if they switch back to tenofovir, these levels decrease again [38], indicating that the effect is specific to tenofovir. However, TAF has not been practiced in Ethiopia.

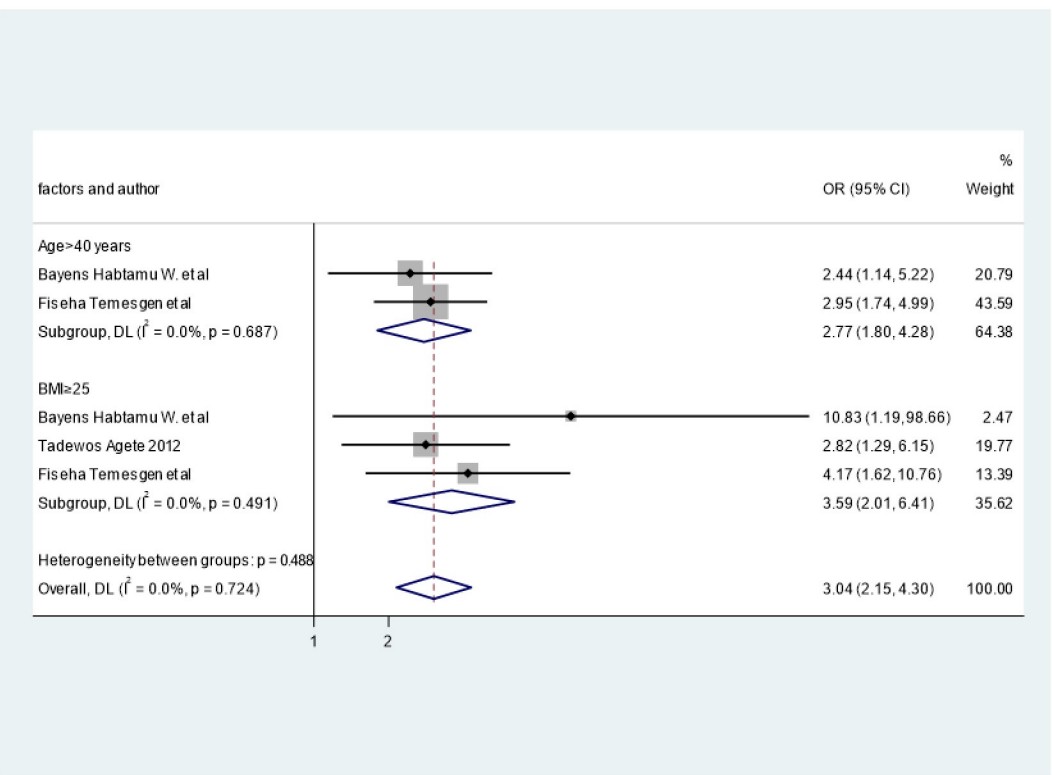

**Fig 8. The effect of age and BMI on elevated levels of TG.**

Again, no significant difference in the prevalence between EFV and NVP-based regimen. Similarly, other studies also found that EFV and NVP is not different in triglyceride levels [39].

The overall pooled prevalence of high LDL-c levels among adult PLHIV was 28.58%. This figure was higher than an African review 23.7% [9]. Specifically, higher among NVP-based regimen 32.03% followed by 30.1% among AZT-based regimen 28.31% among EFV-based regimen,, and 27.17% among D4T-based regimen.

The overall pooled prevalence of low HDL-c levels was 36.17%. This figure was higher than an African review 45.6% [9]. This result was similar with other studies conducted in Zambia 40% [40]. However, it was lower than a study conducted in Spain 44.74% [41]. Even if, no difference found in the prevalence of low HDL-c in EFV and NVP based regimen (36.44% Vs 29.8%, p = 0.6), higher prevalence was observed in the EFV based regimen. Several other studies also found a beneficial effect of NVP on HDL-c [39]. Again, no difference found in the prevalence of low HDL-c among D4T, AZT, and TDF-based regimens (36.1% Vs 32.84%, and 26.46%, p = 0.72). However, the prevalence was high among D4T users.

The possible mechanism of ART induced lipid alteration is inhibition of the mitochondrial DNA (mtDNA) polymerase-γ transcription, leading to mtDNA depletion, which causes diverse pathologies, including lipodystrophy and hepatosteatosis mediated by pro-inflammatory cytokines [42].

There was a strong association between age and high TC. In line with this finding, other studies also reported that increasing age is associated with high TC [43, 44]. Additionally, BMI also associated with high TC. This finding also supported with other study [44]. There was strong association between BMI > 25 and high TG. Similar finding also reported the

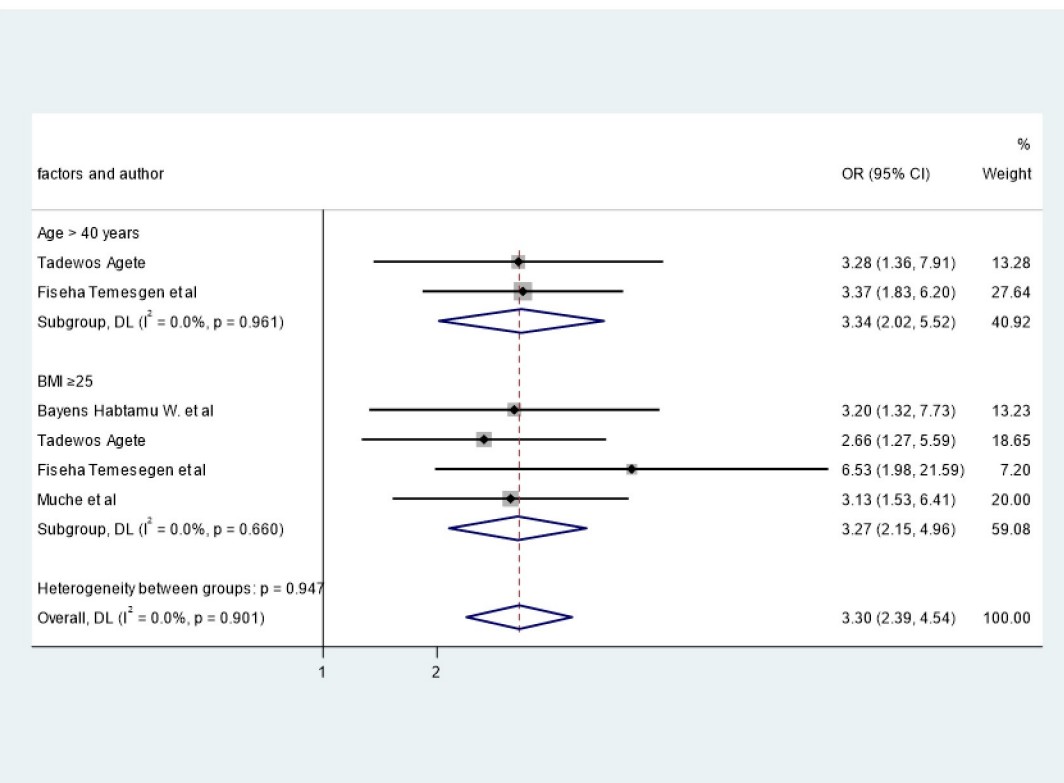

**Fig 9. The effect of age and BMI on LDL.**

association between BMI and High TG [6, 44]. In addition, there are studies in our review that showed the association of CD4 count and TC but we did not show the pooled effect as they used different CD4 count cutoff points [29, 35]. Gebrie A et al. revealed the significant association of substance use, and concomitant drug use for other diseases with dyslipidemia [24]. Nonetheless, as this is the only single study that reported such association, the pooled effect was not shown. Moreover, there was no studies in our review that reported the association of dyslipidemia or any lipid abnormalities with viral load, and opportunistic infections. But longitudinal studies in China showed the association of viral load greater than $10^5$ copies/milliliter with any form of dyslipidemia [45]. All in all, our finding reminds clinicians to make stronger dyslipidemia programs to encounter the growing challenge of CVD. Furthermore, continuous monitoring of dyslipidemia as well as its subclass of lipid abnormalities in PLHIV receiving ART may help recognize those that would have greatest advantage from adjustable factors, such as lifestyle changes, which is associated with dyslipidemia, mainly in older age.

Consider the potential impact on public health and the need for interventions or further research: dyslipidemia is a known risk factor for CVD, which is the leading cause of morbidity and mortality among PLHIV receiving ART. Therefore, effective management of dyslipidemia is important for improving the long-term health outcomes of this population. Currently, in Ethiopia, no local clinical practice guidelines exist for dyslipidemia. Therefore, suitable guidelines should be established to prevent or control dyslipidemia throughout the country. Therefore, PLHIV receiving ART could have continuous monitoring of dyslipidemia for timely preventive or corrective measures.

This review and meta-analysis has implications for clinical practice. Assessing the nationwide prevalence of dyslipidemia in adult PLHIV taking ART would be important to know data on the national level. This data provides for policymakers to implement appropriate measurement. Finally, it is important to establish to prevent CVD as early as possible. In addition, age and BMI were found significant association with high TC, TG and LDL-c. Thus, promoting healthy lifestyles measures could be integrated in prevention programs of dyslipidemia to promote awareness of dyslipidemia among healthcare workers and patients. Furthermore, our finding gives data about the prevalence of dyslipidemia in the county level for possible consideration throughout usual clinical practice.

Our findings have been some limitations. First, we have limited paper. Second, it may need more national representativeness because data were not available from all regions, which could affect the generalizability of our findings. Third, cause-effect relationship not established because of almost all studies were cross sectional. Fourth, in most studies, lipid profiles were measured only once, which may lead to misdiagnosis. Fifth, this review did not analyze some important parameters such as TC/HDL-c and non-HDL-c. Additionally, we were unable to estimate prevalence of dyslipidemia on each ART regimen. Lastly, almost all included studies did not provide age-and sex-specific, and association with viral load and opportunistic infection; hence, we were unable to estimate pooled prevalence of dyslipidemia according to those important characteristics.

## Conclusion

The present systematic review and meta-analysis found that the prevalence of dyslipidemia is high among PLHIV receiving ART. Age and BMI were associated with TC, TG and LDL-c. Thus, PLHIV receiving ART should be monitored their lipid profiles routinely and suitable guidelines should be established to prevent or control dyslipidemia all over the country.

## Supporting information

**S1 Table. PRISMA 2020 checklist.** The updated protocol was followed as a guideline.
(DOCX)

**S2 Table. Quality evaluation of dyslipidemia.** New Castel Ottawa was used.
(DOCX)

**S3 Table. Risk of bias evaluation.** Hoy score was used.
(DOCX)

**S4 Table. Article searching history from different databases.**
(DOCX)

## Author Contributions

**Conceptualization:** Abebe Muche Belete, Daniel Molla Melese, Adisu Asefa, Taklo Simeneh Yazie.

**Data curation:** Wondimeneh Shibabaw Shiferaw.

**Formal analysis:** Abebe Muche Belete, Wondimeneh Shibabaw Shiferaw, Yared Asmare Aynalem, Taklo Simeneh Yazie.

**Methodology:** Bekalu Bewket, Wondimeneh Shibabaw Shiferaw, Yared Asmare Aynalem.

**Validation:** Abebe Muche Belete.

**Visualization:** Belachew Tegegne.

**Writing – original draft:** Abebe Muche Belete.

**Writing – review & editing:** Abebe Muche Belete, Belachew Tegegne, Wondimeneh Shibabaw Shiferaw, Adisu Asefa.

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
