## [Decision Letter · Decision Letter 0]

9 Jan 2024

PONE-D-23-28119Dyslipidemia among adult HIV patients on antiretroviral therapy and its association with age and body mass index in Ethiopia: A Systematic Review and meta-analysis.PLOS ONE

Dear Dr. Belete,

Thank you for submitting your manuscript to PLOS ONE. After careful consideration, we feel that it has merit but does not fully meet PLOS ONE’s publication criteria as it currently stands. Therefore, we invite you to submit a revised version of the manuscript that addresses the points raised during the review process.

We look forward to receiving your revised manuscript.

Kind regards,

Michael Onyebuchi Iroezindu

Academic Editor

PLOS ONE

Reviewers' comments:

Reviewer's Responses to Questions

**Comments to the Author**

1. Is the manuscript technically sound, and do the data support the conclusions?

Reviewer #1: Partly

2. Has the statistical analysis been performed appropriately and rigorously? 

Reviewer #1: Yes

3. Have the authors made all data underlying the findings in their manuscript fully available?

Reviewer #1: Yes

4. Is the manuscript presented in an intelligible fashion and written in standard English?

Reviewer #1: Yes

5. Review Comments to the Author

Reviewer #1: I have read this manuscript with huge interest. Dyslipidaemia contributes to the huge cardiovascular burden in HIV seropositive patients. The authors attempted to carry out a review and meta-analysis of local Ethiopian data on dyslipidaemia in HIV positive patients.

HIV infection is a multi-system disease and the presence of opportunistic infections, immunological status (typified by CD4 count levels), viral load, drugs (both HAART and medications used to treat other illnesses) add to the burden of the disease. The authors described local dyslipidaemia prevalence rates and how HAART affect lipid fractions. They also found that dyslipidaemia had significant relationships with age and BMI.

The authors are advised to relate dyslipidaemia with co-existing opportunistic illnesses, other drugs (apart from HAART which affect lipids), CD4 count and viral load from the available data as this will add more depth to the write up.

There are also a lot of grammatical and syntax errors in the manuscript which the authors should address.

These changes should be made and the manuscript resubmitted.

6. PLOS authors have the option to publish the peer review history of their article (what does this mean?). If published, this will include your full peer review and any attached files.

Reviewer #1: No

---

## [Author Response · Author response to Decision Letter 0]

25 Jan 2024

PLOS ONE Decision: Revision required [PONE-D-23-28119] - [EMID:eae76a4b619b03df]

PONE-D-23-28119

Dyslipidemia among adult HIV patients on antiretroviral therapy and its association with age and body mass index in Ethiopia: A Systematic Review and meta-analysis.

PLOS ONE

Author response to Reviewer and Academic Editor’s comments

Dear Academic Editor and reviewer, we would like to thank you for all your effort and invaluable comments to improve the manuscript. We have done all the best to address both the Academic Editor and reviewer’s comments. All revisions were made as per the comments given. All changes have been highlighted in the manuscript labelled as “Revised Manuscript with Track Changes”.

 We thank you all again!

Author response: Dear Editor, we have read the journal requirements via the links given above and we have made amendments that have been highlighted in the “revised manuscript with track changes”.

Author response: Dear Editor, we thank again for the comment. All data used for this systematic review and meta-analysis are available within the manuscript and /or supplementary file. Therefore, the data availability statement in the online form and in the manuscript is “All data generated and analysed during this study are included in this published article (and its supplementary file)”. 

Reviewers' comments:

Reviewer's Responses to Questions

Comments to the Author

1. Is the manuscript technically sound, and do the data support the conclusions?

Reviewer #1: Partly

Author response #1: Dear reviewer, we made revision to the conclusion and the change has been highlighted in the “revised manuscript with track changes”

2. Has the statistical analysis been performed appropriately and rigorously?

Reviewer #1: Yes

Author response#2: Thank you 

3. Have the authors made all data underlying the findings in their manuscript fully available?

Reviewer #1: Yes

Author response#3: Thank you again for your suggestion.

4. Is the manuscript presented in an intelligible fashion and written in standard English?

Reviewer #1: Yes

Author response#4: Thank you again

5. Review Comments to the Author

Reviewer #1: I have read this manuscript with huge interest. Dyslipidaemia contributes to the huge cardiovascular burden in HIV seropositive patients. The authors attempted to carry out a review and meta-analysis of local Ethiopian data on dyslipidaemia in HIV positive patients.

HIV infection is a multi-system disease and the presence of opportunistic infections, immunological status (typified by CD4 count levels), viral load, drugs (both HAART and medications used to treat other illnesses) add to the burden of the disease. The authors described local dyslipidaemia prevalence rates and how HAART affect lipid fractions. They also found that dyslipidaemia had significant relationships with age and BMI.

The authors are advised to relate dyslipidaemia with co-existing opportunistic illnesses, other drugs (apart from HAART which affect lipids), CD4 count and viral load from the available data as this will add more depth to the write up.

Author response#5: Dear reviewer, your insightful comments have been instrumental in refining our work.

The relationship between viral load and dyslipidemia among people living with HIV/AIDS (PLWHA) who are on highly active antiretroviral therapy (HAART) is a complex and multifaceted one. High viral load is associated with chronic inflammation in the body, which can contribute to dyslipidemia by decreasing lipoprotein lipase activity and increasing cholesterol synthesis. HIV replication activates the immune system, leading to the production of inflammatory cytokines like C-reactive protein (CRP) and interleukin-6 (IL-6). These markers can further interfere with lipid metabolism and promote dyslipidemia.

In response to your suggestions, we have made revisions on CD4, non-ART medication, substance use and comorbidities in the result and discussion sections to provide a more comprehensive description of dyslipidemia. We found that dyslipidemia was associated with low CD4 count, substance use and comorbidity like cancer. While no studies reported on the association between dyslipidemia with opportunistic infection and viral load. We have explicitly addressed this limitation in our manuscript, highlighting the absence of relevant studies on the association between dyslipidemia with opportunistic infections and viral load. Kindly see the revised manuscript.

There are also a lot of grammatical and syntax errors in the manuscript which the authors should address.

Author response#6: Dear reviewer, we had read the whole manuscript and made necessary changes that highlighted in the “revised manuscript with track changes”.

These changes should be made and the manuscript resubmitted.

Author response#6: Dear reviewer, necessary revisions have been made as per your comments and changes have been highlighted in the “revised manuscript with track changes”

---

## [Editor Report · Decision Letter 1]

26 Jan 2024

Dyslipidemia among adult HIV patients on antiretroviral therapy and its association with age and body mass index in Ethiopia: A Systematic Review and meta-analysis.

PONE-D-23-28119R1

Dear Dr. Abebe Muche Belete,

We’re pleased to inform you that your manuscript entitled "Dyslipidemia among adult HIV patients on antiretroviral therapy and its association with age and body mass index in Ethiopia: A Systematic Review and meta-analysis" has been judged scientifically suitable for publication and will be formally accepted for publication once it meets all outstanding technical requirements.

Kind regards,

Michael Onyebuchi Iroezindu

Academic Editor

PLOS ONE
---

## [Editor Report · Acceptance letter]

19 Mar 2024

PONE-D-23-28119R1 

PLOS ONE

Dear Dr. Belete, 

I'm pleased to inform you that your manuscript has been deemed suitable for publication in PLOS ONE. Congratulations! Your manuscript is now being handed over to our production team.

Kind regards, 

on behalf of

Dr. Michael Onyebuchi Iroezindu 

Academic Editor

PLOS ONE